# Long Non-Coding RNAs as Regulators for Targeting Breast Cancer Stem Cells and Tumor Immune Microenvironment: Biological Properties and Therapeutic Potential

**DOI:** 10.3390/cancers16020290

**Published:** 2024-01-10

**Authors:** Fang Yang, Yiqi Yang, Yuling Qiu, Lin Tang, Li Xie, Xiaoxiang Guan

**Affiliations:** 1The Comprehensive Cancer Center of Nanjing Drum Tower Hospital, Affiliated Hospital of Medical School, Nanjing University, Nanjing 210008, China; yangfangnju@hotmail.com (F.Y.); yangyiqi@njmu.edu.cn (Y.Y.); yuling_qiu@smail.nju.edu.cn (Y.Q.); 2Clinical Cancer Institute, Nanjing University, Nanjing 210008, China; 3Department of Rheumatology and Immunology, Jinling Hospital, Affiliated Hospital of Medical School, Nanjing University, Nanjing 210002, China; njutanglin@hotmail.com; 4Department of Oncology, The First Affiliated Hospital of Nanjing Medical University, Nanjing 210029, China

**Keywords:** lncRNA, breast cancer stem cell, heterogeneity, tumor immune microenvironment, therapeutic target

## Abstract

**Simple Summary:**

A number of lncRNAs participate in the stemness-related signaling pathways. In this article, we provided insights into the emerging data for the interaction between BCSCs and TIME, and comprehensively reviewed the functions and mechanisms of lncRNAs in maintaining stemness and reprogramming TIME in breast cancer. The key role of lncRNAs in maintaining stem-like properties and tumor immune microenvironment implied that lncRNAs can be employed as potential therapeutic targets for breast cancer, as well as biomarkers for diagnosis and prognostic evaluation.

**Abstract:**

Breast cancer stem cells (BCSCs) is a subpopulation of cancer cells with self-renewal and differentiation capacity, have been suggested to give rise to tumor heterogeneity and biologically aggressive behavior. Accumulating evidence has shown that BCSCs play a fundamental role in tumorigenesis, progression, and recurrence. The development of immunotherapy, primarily represented by programmed cell death protein 1 (PD-1)/programmed death-ligand 1 (PD-L1) inhibitors, has greatly changed the treatment landscape of multiple malignancies. Recent studies have identified pervasive negative associations between cancer stemness and anticancer immunity. Stemness seems to play a causative role in the formation of cold tumor immune microenvironment (TIME). The multiple functions of long non-coding RNAs (lncRNAs) in regulating stemness and immune responses has been recently highlighted in breast cancer. The review focus on lncRNAs and keys pathways involved in the regulation of BCSCs and TIME. Potential clinical applications using lncRNAs as biomarkers or therapies will be discussed.

## 1. Introduction

Breast cancer is a highly heterogeneous disease, which can be classified into four molecular subtypes according to the expression levels of estrogen receptor (ER), progesterone receptor (PR), and human epidermal growth factor receptor 2 (HER2). The unique expression profile of proteins forms the histological basis of each subtype and provides prognostic information. Apart from the heterogeneity across different subtypes (intertumor heterogeneity), tremendous heterogeneity within a single patient (intratumor heterogeneity, ITH) has been highlighted. Furthermore, ITH consists of spatial heterogeneity and temporal heterogeneity, which represent the differences in genetic, phenotypic, or biological behavioral characteristics of the tumors across different geographical regions or at different time points, respectively. The remarkable spatial and temporal ITH in breast cancer have posed obvious impediments to clinical accurate diagnosis and prognosis, as well as development of targeted agents, since a routine single biopsy might misjudge the complexity and evolutionary dynamics of the disease.

Cancer stem cell (CSC) hypothesis is one of the major concepts aiming to interpret the causes of ITH. CSCs have been proposed to contribute to multistep aggressive processes in breast cancer [1,2]. Recent evidence suggests that tumor microenvironment (TME) dictates the CSCs maintenance to arbitrate cancer progression and metastasis. CSCs could impact antitumor immunity by interacting with TME components, including immune cells, endothelial cells, cytokines, chemokines, etc. Thus, targeting the CSCs and tumor immune microenvironment (TIME) may signify an effective therapeutic strategy for drug resistance. 

Long non-coding RNAs (lncRNAs), which are at least 200 nucleotides long and do not encode any protein, are involved in regulating genes at the transcriptional, post-transcriptional, and epigenetic levels. For instance, lncRNAs can modulate chromatin structure and function, regulate the transcription of neighboring and distant genes, and affect the splicing and stability of mRNAs [3,4]. A number of lncRNAs have been found to play important roles in various cancer biological processes, including cell proliferation, invasion, metastasis, and drug resistance [5,6,7,8]. Recent reports have shown that lncRNAs might have an impact on key signaling pathways that govern the cancer hallmarks associated with breast CSCs (BCSCs). LncRNAs are also specifically expressed in a variety of immune cell types and may play crucial roles in TIME through multiple regulatory mechanisms, making them prognostic or predictive biomarkers and attractive targets for therapy. In this review, we provide an overview of the lncRNAs associated with BCSCs and TIME, and further propose rational therapeutic strategies for breast cancer treatment.

## 2. BCSC and TIME

### 2.1. BCSC Markers

A variety of surface biomarkers have been utilized to recognize or isolate BCSCs. The first identified BCSCs were a subset of cells with CD44^+^CD24^−/low^ phenotype, which could efficiently form tumors in NOD/SCID mice, whereas tens of thousands of CD44^−^/CD24^+^ cancer cells could not [9]. ALDHA1 and CD133 are another two acknowledged markers of BCSCs. ADELFLUOR assay can effectively identify cells with ALDH1 activity. ALDEFLUOR-positive breast cancer cells display properties of cancer stem cells, such as self-renew, generating ALDEFLUOR-positive cells, and even differentiating to ALDEFLUOR-negative cells [10]. CD133^+^ cells were found to have stem-like properties such as self-renewal and differentiation capabilities in BRCA1-knockout mice [11]. In addition, high expressions of CD49f, CD29, and CD61 are associated with high capacity for self-renewal in breast cancer [12,13]. We have summarized the emerging BCSC markers and their specific expressions in different cell lines or clinical subtypes, which implies that BCSCs are essentially heterogeneous [14]. The diversity of CSCs may attribute to the complexity and variability during the evolutionary processes with the help of driver mutations and the factors in the TIME [15].

### 2.2. Role of TIME in CSCs

#### 2.2.1. Hypoxia

Hypoxia has been found to promote stem-like phenotype in non-stem cell population of glioblastoma by upregulating OCT4, NANOG, and c-MYC, thereby promoting the self-renewal properties of the stem and non-stem cancer cells [16]. In breast cancer, hypoxia also contributes significantly to induce the stem-like phenotype [17]. The exposure of breast cancer cells to hypoxia increased NANOG mRNA and protein expression and induced BCSC phenotype by a hypoxia-inducible factor (HIF)- and ALKBH5-dependent manner [17]. Hypoxia was found to suppress DICER expression, a key enzyme involved in microRNA processing, through an epigenetic mechanism and ultimately promotes stem cell phenotypes and poor prognosis of breast cancer [18]. Moreover, HIFs were required for chemotherapy resistance of BCSCs, suggesting that combining chemotherapy with agents that block HIF activity may be effective therapeutic options in breast cancer patients [19].

#### 2.2.2. Stromal Cell Interactions

Cancer-associated fibroblasts (CAFs) are a predominant component of the TIME and play a fundamental role in tumorigenesis, cancer progression, and drug resistance by sustaining cancer stemness [20]. It has been reported that CAFs can activate Wnt/β-catenin and Notch signaling pathways involved in maintaining stemness in breast cancer [21,22]. In return, CSCs regulate CAF activity via the Hedgehog signaling pathway in mammary gland tumors as a feedback model [23]. Targeting CAFs with inhibitors against Wnt/β-catenin, Notch, or Hedgehog signaling pathways may provide a novel therapeutic strategy against breast cancer. Indeed, CAF contains heterogeneous subclusters with distinct properties in breast cancer [24,25]. The CAF-S1 subpopulation promotes an immunosuppressive environment by attracting and retaining CD4^+^/CD25^+^ T lymphocytes though a multi-step mechanism. CAF-S1 cells also enhance the regulatory T cell differentiation and activity to inhibit T effector proliferation, while CAF-S4 cells do not exhibit these properties [25]. Tumor-associated macrophages (TAMs) constitute another major cell population in the TIME. Co-injection of TAMs with CSCs significantly promoted the growth of breast cancer cells compared with injection of CD44^+^/CD24^−^ cells alone [26]. Further mechanism study implied that HAS2 was crucial for the interaction between CSCs and TAMs, leading to increased secretion of PDGF-BB by TAMs, thereby activating stromal cells and enhancing self-renewal capability of CSCs [26]. TAM-derived IL-6 enhances CSCs phenotype and expression of CSC specific transcription factors including NANOG, SOX2, and OCT3/4, via activating the STAT-3 signaling pathway in breast cancer cells [27]. In addition, TAMs promote the epithelial–mesenchymal transition (EMT) and CSC properties by activating the CCL2/AKT/β-catenin pathway in triple-negative breast cancer (TNBC) [28]. Therefore, it may be a novel therapeutic strategy by influencing macrophage activity or interaction with CSCs for breast cancer patients. Cytokines, chemokines, and growth factors are also critical components of the TIME, regulating stem-like properties and phenotypes through the interactions with CSCs. Targeting the elevated cytokines in the TIME may effectively eradicate CSCs, and inhibit CSC-induced progression of breast cancer [29]. In addition, extracellular matrix (ECM)-derived mechanical force has recently been found to regulate breast cancer stemness and cell quiescence [30]. Moreover, three-dimensional (3D) culture system of biomimetic scaffolds could restate the biophysical characteristic of ECM and enhance the stemness properties of esophageal squamous cell carcinoma cells both in vitro and in vitro [31]. Taken together, CSCs are regulated by complex interactions with the various components of the TIME, such as stromal cells, immune cells, immunomodulatory factors, and ECM, which represent potential targets for therapies in the future.

## 3. Biological Characteristics and Mechanisms of LncRNAs

Initially, lncRNAs were considered as byproducts of RNA polymerase II which did not have biological functions. However, recent studies have shown that lncRNAs have conserved secondary structures that can interact with proteins, DNA, and mRNA, and participate in the regulation of multiple biological processes of cancers. LncRNAs can be classified into five categories according to their genomic proximity to neighboring transcripts [32,33,34]: (i) antisense lncRNAs, which overlap with one or more exons of the coding gene on the opposite strand; (ii) enhancer lncRNAs, which are produced by the enhancer region of the coding gene; (iii) intergenic lncRNAs, also known as lincRNA, which are independently transcribed by sequences located between the coding genes; (iv) bidirectional lncRNAs, which share the same promoter with the coding gene, but are transcribed in the opposite direction; (v) intronic lncRNAs, which are produced by the intron region of the coding gene.

LncRNAs can regulate gene expression at numerous levels by various mechanisms: (i) Transcriptional interference: transcription from an upstream noncoding promoter to negatively regulate the expressions of downstream genes. For example, lncRNA SRG1 transcription derepresses SER3 by directing a high level of nucleosomes over SRG1, which overlaps the SER3 promoter [35]. (ii) Induction of chromatin remodeling and histone modifications: inhibiting RNA polymerase II or inducing chromatin remodeling and histone modification to positively affect the expressions of downstream genes. For example, lncRNA TGFB2-AS1 interacts with chromatin remodeling complex SWI/SNF and results in transcriptional repression of its target genes including TGFB2 and SOX2 [36]. LncRNA HOTAIR acts as a scaffold to mediate the interaction with histone modification complexes, such as PRC2 and LSD1, enabling the assembly of select histone modification enzymes [37]. (iii) Modulation of alternative splicing patterns: forming complementary double-stranded complexes with transcripts of encoding gene, interfering with splicing of mRNAs, and generating different splicing forms. For example, lncRNA MALAT1 modulates alternative splicing by cooperating with the splicing factors PTBP1 and PSF [38]. (iv) Generation of endo-siRNAs: production of endo-siRNAs by Dicer enzymes. For example, RMRP, a 267-nucleotide noncoding RNA, can generate endo-siRNAs depending on Dicer [39]. (v) Structural or organizational role: acting as a structural component to form a RNA–protein complex. For example, Xist is a lncRNA essential for X-chromosome inactivation. The Xist RNA structure can modulate protein interactions through multiple mechanisms [40]. (vi) Modulation of protein activity, and (vii) alteration of protein localization by binding to specific protein transcripts. For example, lncRNA PRADX suppresses UBXN1 expression by recruiting PRC2/DDX5 complex and promotes the nuclear transport of NF-κB [41]. (viii) Small RNA precursor: serves as a precursor of small RNA, such as miRNA and piRNA. For example, the production of DNA damage-induced lncRNAs by Pol II at DNA breaks serves as precursors for small RNAs and recruiters of 53BP1 for DNA repair [42].

## 4. LncRNAs and Regulatory Pathways Associated with BCSC

### 4.1. TGF-β/Smad Signaling Pathway

It is known that the TGF-β/Smad signaling pathway plays an inimitable role in promoting EMT, stemness, and metastasis [43]. LncRNA ROR, a key regulator of self-renewal and differentiation, has recently been found to promote tumor growth and invasion by regulating the TGF-β signaling pathway in breast cancer [44]. LncRNA Smyca was associated with inferior outcomes of multiple cancer types. Mechanistically, Smyca can enhance TGF-β/Smad signaling pathway via acting as a scaffold for enhancing Smad3/Smad4 complex formation, and then potentiate stemness features [45]. LncRNA TGFB2-antisense RNA1 (TGFB2-AS1) has been identified as a critical regulator for the plasticity and reversibility of noncancer stem cells, by suppressing both TGFβ2 and CSC signaling and the downstream targets in TNBC [36]. TGF-β was found to upregulate AC026904.1 and UCA1, two lncRNAs highly expressed in advanced breast cancer, via Smad and ERK pathways, respectively [46]. LncRNA LITATS1 was also a key determinant of epithelial integrity maintenance, which could inhibit TGF-β-induced EMT in both breast and non-small cell lung cancer cells [47].

### 4.2. Hippo/YAP Signaling Pathway

The Hippo signaling pathway has been identified as a tumor suppressor pathway closely involved in regulating the stem-like features in breast cancer [48]. TAZ and YAP are core transcriptional coactivators of the Hippo pathway. Further studies showed that TAZ/YAP activity played a causal role in promoting CSC features and was necessary to maintain self-renewal capacity in breast cancer [49]. In liver cancer, lncRNA NEAT1 was involved in enhancing self-renewal and tumor-initiation capacities through the Hippo signaling pathway [50]. In breast cancer, lncRNA lncROPM was required for sustaining BCSC traits by upregulating its target PLA2G16 expression, which activated Hippo/YAP, Wnt/β-catenin, and PI3K/AKT pathways [51]. LncRNA SOX21-AS1 was identified to attenuate the Hippo signaling activity by promoting YAP nuclear localization, thereby enhancing the stemness of breast cancer cells [52].

### 4.3. Hedgehog Signaling Pathway

The Hedgehog (Hh) signaling pathway has a critical role in maintaining stem-like properties. The oncogenic lncRNA DUXAP10 was found significantly upregulated in Cadmium (Cd)-transformed cells, which displayed CSC-like properties. The knockdown of DUXAP10 of Cd-transformed cells inactivated the Hedgehog pathway by decreasing the expressions of GLI1, SHH, and PTCH2, thereby reducing the CSC-like properties [53]. The SHH-GLI1 pathway related lncRNA-Hh can strengthen CSC generation in breast cancer via activating the Hedgehog signaling and increasing the expressions of downstream targets SOX2 and OCT4 [54]. Some preclinical agents inhibiting the Hedgehog pathway has been discovered, such as cyclopamine and robotnikinin [55]. However, none of them have been applied in clinical practice due to the lack of effective validation of safety and efficacy.

### 4.4. Other Signaling Pathways

Notch signaling pathway plays a crucial role in cell fate determination by regulating proliferation, differentiation, and apoptosis [14]. Activated Notch signaling promotes BCSC characteristics including sphere formation and tumor initiation capacity in breast cancer cells [56]. LncRNA CCAT2 were found overexpressed in TNBC and positively regulated the stemness features of TNBC cells via upregulating OCT4-PG1 expression and activating Notch signaling [57]. The Wnt/β-catenin signaling pathway also plays a substantial role in maintaining BCSC stemness, as well as promoting BCSC proliferation, migration, and invasion. LncRNA LncCCAT1 can enhance BCSC properties by functioning as a molecular sponge for miR-204/211, miR-148a/152, and ANXA2, thus upregulating TCF4 or facilitating β-catenin translocation into the nucleus, leading to the activation of Wnt/β-catenin signaling [58]. LncRNA LUCAT1 can also increase BCSC functions through the LUCAT1/miR-5582-3p/TCF7L2 axis via activating the Wnt/β-catenin signaling pathway [59]. A variety of NF-κB-regulated lncRNAs have been reported to regulate stemness in breast cancer. LncRNA HOTAIR is a non-coding RNA that regulates various genes and signaling pathways related to breast cancer development, metastasis, and drug resistance [60]. Recently, HOTAIR was also found to be required for self-renewal of BCSCs and tumor propagation. Mechanistically, HOTAIR inhibits IκBα expression by recruiting the PRC2 protein complex to the promoter, thereby triggering the NF-κB signaling pathway [61]. Taken together, a number of lncRNAs participate in the stemness-related signaling pathways, which can serve as potential therapeutic targets for the elimination of BCSCs (Figure 1).

## 5. Immune-Specific LncRNAs in the TIME

### 5.1. CAFs

CAFs are the most abundant cell component in the TIME that involved in proliferation, invasion, and anticancer therapeutic resistance [62]. Unlike normal fibroblasts, CAFs exhibit contractility and actively remodel the structure and composition of ECM, and thereby stimulate tumor progression [63]. Recent studies of CAF heterogeneity indicate that multiple CAF subtypes coexist in the TIME, playing distinct roles in promoting tumor growth and modulating therapy responses. It seems that different CAF subtypes are associated with certain subtypes of breast cancer. Costa et al. established four CAF subpopulations (CAF-S1, CAF-S2, CAF-S3, and CAF-S4) according to the expression levels of six fibroblast markers. They found that most luminal A tumors were enriched in CAF-S2 cells, HER2 in CAF-S4, and TNBC either in CAF-S1 or in CAF-S4. CAF-S1 subset correlates with an immunosuppressive microenvironment through a multi-step mechanism [25]. Abnormal expression of lncRNAs may mediate the transformation of normal fibroblasts to CAFs, which promote tumor growth and modulate immune responses [64]. Li et al. identified a number of CAF-specific lncRNAs (FibLnc) based on immune cell transcriptome expression profiling. They found that the high FibLnc score correlated with worse overall survival (OS) in breast cancer patients. Moreover, the FibLnc score was positively associated with immune cell dysregulation score and stronger immune response to anti-PD1 and anti-CTLA4 therapy [65].

### 5.2. TAMs

TAMs are usually classified into two subgroups according to different functions, namely classical activated macrophages (M1) and alternating activated macrophages (M2). M1 macrophages mainly participate in pro-inflammatory response and anti-tumor immunity, while M2 macrophages result in anti-inflammatory response and pro-tumor activities [66]. Increasing evidence indicates that lncRNAs may be involved in the regulation of macrophage polarization. LncRNA BCRT1 was remarkably upregulated in breast cancer tissues and was associated with shorter disease-free survival and OS in breast cancer patients. Mechanistically, lncRNA BCRT1 could promote M2 polarization by competitively binding with miR-1303 to prevent the degradation of its target gene PTBP3, which further enhanced breast cancer progression [67]. Similarly, lncRNA XIST could mediate M2 macrophage polarization by competing with miR-101 to upregulate C/EBPα and KLF6 expression in breast cancer [68]. LncRNA GNAS-AS1 facilitated progression of ER+ breast cancer cells by promoting M2 macrophage polarization via directly sponging miR-433-3p, and subsequently upregulated its target GATA3 [69]. LncRNA LINC00514 has been reported to participate in polarizing macrophages to the M2 phenotype by activating Jagged1-mediated Notch signaling pathway via increasing phosphorylation of STAT3 [70].

### 5.3. Myeloid-Derived Suppressor Cells (MDSCs)

MDSCs are a heterogeneous cell subset comprising mononuclear and polymorphonuclear myeloid cells, with immune suppressive properties in the TIME [71]. They have potent mechanisms to inhibit T-cell and NK-cell activity, thereby negatively affect immune response and contribute to resistance to immunotherapy [72]. Recent studies have shown that lncRNAs may be involved in regulating the immunosuppressive activity of MDSCs. Lnc-C/EBPβ is a novel lncRNA identified in MDSC that can negatively regulate the immunosuppressive function of MDSCs. Mechanistically, inflammatory environment induces lnc-C/EBPβ expression, which binds to the C/EBP isoform liver-enriched inhibitory protein (LIP) to hinder the activation of C/EBP genes [73]. Adewunmi et al. explored the connection between the lncRNA MALAT1 and the TIME by TNBC mouse models. They found that inhibiting MALAT1 with MALAT1 ASO decreased immunosuppressive functions of TAMs and MDSCs, while increased CD8+ T-cell infiltration in the TIME. Additionally, combination of MALAT1 ASO with chemotherapy or immune checkpoint inhibitor significantly improved the treatment responses in preclinical mouse models [74].

### 5.4. Regulatory T Cells (Tregs)

Tregs are a subset of CD4+ T cells that have significant immunosuppressive effects in the TIME [75]. Tumor-infiltrating Tregs could control the autoimmune responsiveness of the body by inhibiting tumor-specific immune effector cells. A number of lncRNAs have been found that display the abnormal expression patterns in Tregs and can participate in the regulation of Tregs functions in breast cancer. Expressions of four Tregs-related lncRNAs, namely RMRP, TH2-LCR, MAFTRR and GATA3-AS1, were significantly higher in breast cancer samples compared to adjacent normal tissues [76]. An exosome-related lncRNAs risk model was established based on the public RNA-sequencing data from TCGA which could predict survival outcomes in breast cancer patients. In addition, there was a significant difference in the infiltration levels of Tregs between the different risk groups, and the exosome-related lncRNAs risk model could efficiently predict the response of immunotherapy in breast cancer patients [77]. CD73+γδ1 T cell was identified as a predominant subpopulation of Tregs exerting immunosuppressive functions in breast cancer. Breast cancer-derived exosomes could induce the expression of CD73 in γδ1 T cell by delivering lncRNA SNHG16. Furthermore, SNHG16 acted as a ceRNA by sponging miR-16-5p to upregulate the expression of SMAD5, which activated the TGF-β1/SMAD5 signaling pathway [78].

### 5.5. Natural Killer Cells (NKs)

NKs belong to the first line of immune defense, which can directly kill cancer cells by secreting perforin, IFN-γ, TNF, and GM-CSF in the TIME [79]. The dysregulated lncRNAs in NKs were closely associated with the cytotoxicity and differentiation of NKs. LncRNA TYMSOS has been shown to promote immune escape and repress the NK92 cells cytotoxicity in breast cancer [80]. Knocking down of lncRNA MALAT1 significantly boosted NK cells-mediated killing and CD8+ T cells-mediated cytotoxicity in TNBC cells [81]. LncRNA UCA1 was found to enhance the expression of ULBP2 and promote the detachment of soluble ULBP2 from the cell surface, making it resistant to NK cells-mediated killing in breast cancer [82]. These studies fully demonstrated the potential and importance of lncRNAs in regulating the functions of NKs and modulating the anti-tumor immune response.

Taken together, the TIME is mainly composed of cancer cells, stromal cells, immune cells, microvessels, and various cytokines and chemokines. The multiple lncRNAs derived from TIME can be involved in mediating immune and cancer cell interactions and thereby affect the immune response (Figure 2).

## 6. Clinical Implications of lncRNAs in BCSC and TIME

### 6.1. Prognostic Value of Stemness-Related LncRNAs

Increasing evidence indicates that lncRNAs are implicated in the regulation of stemness and are closely associated with the prognosis of breast cancer patients (Table 1) [36,51,83,84,85,86,87,88,89]. TGFB2-AS1 impaired the CSC properties and dramatically inhibited the malignant character of TNBC cells, and high TGFB2-AS1 expression was correlated with a better outcome in TNBC patients [36]. LncRNA FGF13-AS1 has been identified as a tumor suppressor by attenuating stemness properties of breast cancer cells, and the decreased FGF13-AS1 levels was correlated with poor prognosis [83]. On the contrary, LncROPM was found to be highly expressed in BCSCs and was positively correlated with poor prognosis in breast cancer patients [51]. LncRNA KB-1980E6.3, a novel lncRNA aberrantly upregulated in breast cancers, has been proven to enhance self-renewal ability in vitro and be closely associated with the inferior prognosis of breast cancer patients [84]. A previous study identified 12 BCSC-related lncRNAs with significant prognostic value by an optimal prognostic risk model. The risk model was further verified as an independent prognostic factor for breast cancer patients. The stemness statuses were quite different between the high-risk and low-risk groups. The high-risk group was obviously enriched in stemness-related pathways and oncogenic signatures, and presented a significantly worse OS than the low-risk group [90]. Moreover, whether the lncRNAs have a superior prognostic value to other clinicopathological factors (e.g., tumor size, lymph node status, TNM stage) remains unclear.

Notably, although a number of lncRNAs have been suggested to have potential prognostic value in breast cancer, most of them were investigated in early preclinical stages by limited sample sizes. Further studies of large-scale clinical validation are needed to confirm the reliability and accuracy of lncRNAs as prognostic markers for breast cancer.

### 6.2. LncRNAs Impact Sensitivity to Immunotherapy

Immune checkpoint inhibitors (ICIs) targeting PD-1/PD-L1 and CTLA-4 have greatly changed the landscape of cancer therapy. However, only a subset of patients can derive meaningful clinical responses and benefits from the immunotherapy, and most patients develop secondary resistance. The dysregulations in the TIME, including abnormal metabolism, defective antigen presentation, and the recruitment of immunosuppressive cells and cytokines, lead to the formation of an immunosuppressive microenvironment, which facilitates immune escape and tumor invasion [91]. LncRNAs play an essential role in the regulation of metabolic reprogramming and immune microenvironment remodeling, and promote the resistance to immunotherapy in breast cancer. LncRNA GATA3-AS1 facilitates tumor progression and immune evasion in TNBC by destabilization of GATA3 and stabilization of PD-L1 [92]. LncRNA TINCR impairs the efficacy of PD-L1 inhibitor in breast cancer by recruiting DNMT1 and functioning as a sponge of miR-199a-5p and upregulating the stability of USP20 mRNA. The knockdown of TINCR significantly enhances PD-L1 inhibitor sensitivity in breast cancer both in vitro and in vivo, producing a synergistic anticancer effect [93]. LINK-A was demonstrated to downregulate antigen presentation through inactivating the PKA pathway, and a combination of LINK-A locked nucleic acids (LNAs) and ICIs displayed a synergistic anticancer effect in TNBC [94]. However, these explorations are still in the preclinical stage, and more clinical studies are needed to verify the relationship between lncRNAs and immunotherapy responses.

### 6.3. Strategies of Eliminating CSCs and Regulating TIME by Targeting LncRNAs

Since lncRNAs function as key players in maintaining stem-like properties and TIME remodeling, suppressing the stem phenotype and reversing the immunosuppressive microenvironment by targeting lncRNAs may provide a potential therapeutic idea for the treatment of breast cancer. Hypoxia-induced lncRNA H19 was found to promote glycolysis and stemness by elevating PDK1 expression in breast cancer. Interestingly, aspirin remarkably inhibited glycolysis and stem-like features by dampening expressions of H19 and PDK1. The preclinical experiments revealed that aspirin downregulated the expressions of stemness-related genes and decreased the numbers and diameters of mammospheres in vitro [95]. Notably, an increasing number of studies have reported that aspirin can exert an inhibitory effect on BCSCs as well as the traditional anti-inflammatory activity. Zhang et al. found that aspirin could decrease the expressions of CSC markers and attenuate self-renewal capacity in pancreatic cancer cells, which meanwhile did not exhibit obvious cytotoxicity on normal cells [96]. Guo et al. found that aspirin exhibited anti-metastatic activity in lung metastasis of colorectal cancer by transcriptionally activating lncRNA OLA1P2 expression, which inhibited STAT3 signaling pathway activity [97]. Currently, many natural compounds including tangeretin, CaA, glabridin, and plumbagin were also demonstrated to be able to eliminate stem-like properties in breast cancer [98,99,100,101]. However, the role of lncRNAs in these drug-induced reduction in stemness properties remains unknown. It is attractive to define the therapeutic efficacy of chemical drugs or natural compounds against lncRNAs in BCSCs in future studies.

Nucleic acid-based technologies have recently been developed to silence lncRNAs. The traditional method for silencing RNA, known as RNA interference (RNAi), has been demonstrated to be inefficient for lncRNAs located in the nucleus. The Antisense Oligonucleotide (ASO) Gapmer contains a central DNA portion that promotes RNA degradation mediated by RNase H. Since RNase H can be distributed both in cytoplasm and nucleus, ASO Gapmer-based knockdown is considered to be powerful regardless of the locations for the target lncRNAs [102]. Among the ASOs, locked nucleic acid (LNA) is a specially modified antisense nucleic acid molecule with high affinity and stability in silencing lncRNAs [103,104]. LncRNA LINK-A initiates mammary gland tumors that resemble human TNBC, while LINK-A LNAs therapy can suppress tumor progression and sensitize breast tumors to immunotherapy [94]. LncRNA PKMYT1AR is identified as a human-specific lncRNA that promotes CSC maintenance in non-small cell lung cancer. CSCs self-renewal ability was markedly repressed upon PKMYT1AR ASOs treatment [105]. Taken together, the modification of lncRNAs by ASO treatment may be a promising therapeutic approach for breast cancer patients.

## 7. Conclusions and Future Perspectives

This article mainly reviews the functions and mechanisms of lncRNAs in maintaining stemness and reprogramming the TIME in breast cancer. The key role of lncRNAs in maintaining stem-like properties and tumor immune escape implied that lncRNAs can be employed as potential therapeutic targets for breast cancer, as well as biomarkers for diagnosis and prognostic evaluation. It may achieve a synergistic effect for immunotherapy when combining with the therapies targeting the CSC- or TIME-related lncRNAs. Although lncRNAs have shown great potential in immunotherapy, a clear mechanism still remains to be elucidated. Also, there are some problems that need to be solved before successfully transforming the lncRNA-based therapy into clinical application. First, it is a challenge to develop a safe and effective delivery system to deliver lncRNAs to specific organs or cells. Optimizing the delivery molecules to maximize anti-tumor activity and minimize off-target effects is the core of the design of the delivery system. Second, the TIME is highly heterogenous, consisting of multiple cell populations across different cancer types; meanwhile, the lncRNAs display a strong cell type-specific expression pattern in the TIME [106]. The efficacy of lncRNA-based therapy in a certain patient could not be validated in other patients. In addition, there are no lncRNA-specific animal models available for current cancer research, due to the lack of conserved lncRNA sequences between the experimental animal models and humans. There is an urgent need to establish an animal model that can express human lncRNAs. Organoids may be a potential model for preserving tumor heterogeneity and expressing human lncRNAs. Next, numerous lncRNAs have been discovered that are involved in the regulation of immune responses, but it is difficult to answer which one is paramount to impact the efficacy of immunotherapy. Combining lncRNA-based targeted therapy with conventional chemotherapy or immunotherapy may provide a novel direction for anti-tumor therapy. To date, no clinical trials are available to explore the anti-tumor activity of lncRNA-based targeted therapy. Further studies should shed light on the mechanism of lncRNAs in regulating TIME-mediated stemness, and explore the potential therapeutic approaches targeting lncRNAs to improve response to immunotherapy.

## Figures and Tables

**Figure 1 cancers-16-00290-f001:**
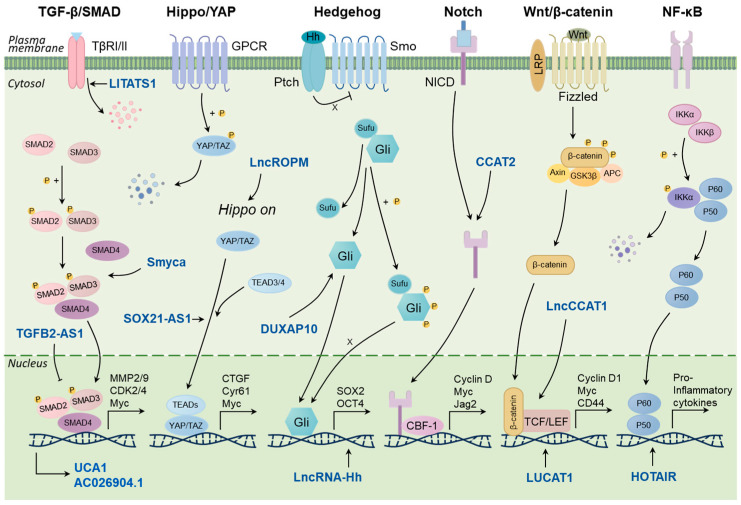
LncRNAs regulate the key signaling pathways related to stemness in breast cancer.

**Figure 2 cancers-16-00290-f002:**
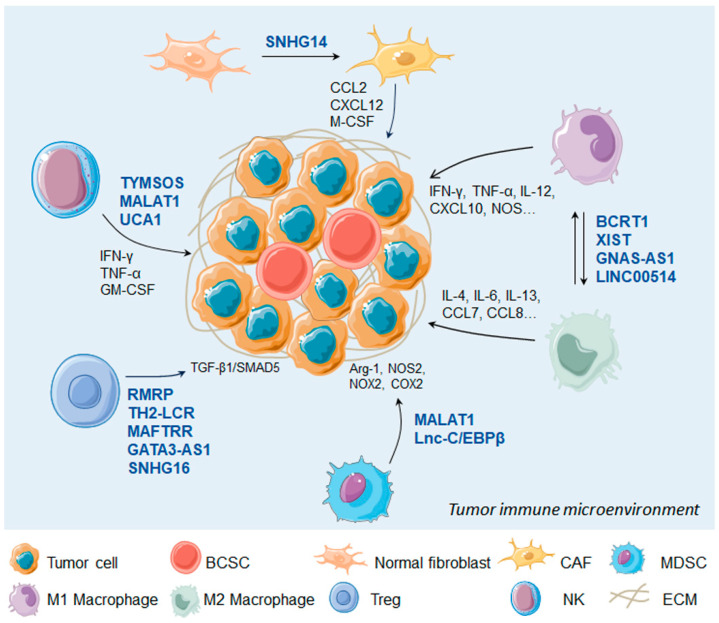
LncRNAs regulate immune cells in the tumor microenvironment. BCSC, breast cancer stem cell; CAF, carcinoma associated fibroblast; MDSC, myeloid-derived suppressor cell; NK, natural kill cell; ECM, extracellular matrix.

**Table 1 cancers-16-00290-t001:** Stemness-related lncRNAs with prognostic value.

LncRNA	Alterations	Targets	Functions	Prognostic Value
TGFB2-AS1 [36]	Downregulated in MDA-231-LM2 cells (showing enhanced lung metastatic activity compared to the parental MDA-231 cells)	TGFβ2, SOX2	Attenuate BCSC self-renewal activity and inhibit the malignant character	Correlated with better outcome
FGF13-AS1 [83]	Downregulated in breast cancer cells	IGF2BPs, MYC	Inhibit breast cancer cells proliferation, invasion, metastasis, glycolysis and stemness features	Correlated with better outcome
LncROPM [51]	Upregulated in BCSCs	PLA2G16	Promote BCSC stemness by regulating PLA2G16-mediated phospholipid metabolism	Correlated with poor outcome
KB-1980E6.3 [84]	Upregulated in hypoxic breast cancer cells	IGF2BP1, c-MYC	Maintain the stemness of BCSCs	Correlated with poor outcome
Lnc030 [85]	Upregulated in BCSCs	SQLE	Maintain BCSC stemness by stabilizing SQLE mRNA and increasing cholesterol synthesis	Correlated with poor outcome
LINC00511 [86]	Upregulated in breast cancer cells	miR-185-3p/E2F1/NANOG	Promote proliferation, invasion, maintenance of BCSC features	Correlated with poor outcome
XIST [87,88]	Downregulated in breast cancer cells	let-7a-2-3p/IL-6/STAT3	Promote BCSCs by activating proinflammatory IL-6/STAT3 signaling	Correlated with poor outcome
SNHG7 [89]	Upregulated in chemoresistant breast cancer tissues and cells	miR-34a	Modulate chemoresistance and stemness partially via MiR-34a	Correlated with chemoresistance

BCSC: breast cancer stem cell.

## Data Availability

The data can be shared up on request.

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
