# Peer review of "Long Non-Coding RNAs as Regulators for Targeting Breast Cancer Stem Cells and Tumor Immune Microenvironment: Biological Properties and Therapeutic Potential"

_cancers, 2024, doi:10.3390/cancers16020290_

Round 1

Reviewer 1 Report

Comments and Suggestions for Authors

I consider it a good article, well structured and precise.

Comments on the Quality of English Language

Minor editing of English language required

Author Response

Comments 1: I consider it a good article, well structured and precise.

Response 1: Thank you very much for the positive comments.

Comments 2: Minor editing of English language required

Response 2: Thank you for your comment. We have made minor editing of our language.

Reviewer 2 Report

Comments and Suggestions for Authors

The manuscript by F. Yang et al. is addressed to the regulation of breast cancer stem cells and their interaction with cells in the tumor microenvironment by long non-coding RNAs.

The data described by the authors is well illustrated; the manuscript contains two figures. The table summarizes published data on the role of long non-coding RNAs in carcinogenesis. Unfortunately, about 20% of the data is taken by the authors from reviews rather than from original research articles, which reduces the value of the material.

Major comment.

The authors are encouraged to include more references to original research articles and reduce the number of references to reviews.

Lines 162-173. To confirm the data presented in this paragraph, a reference to another review article was used. I would ask the authors to provide references to original research articles confirming the functions of lncRNAs for each designated function (i-viii).

Minor comments.

Line 88. Probably a typo: BSCSs instead of BCSCs.

Line 151. Better: Initially, lncRNA was considered as a byproduct of RNA polymerase II which did not have biological functions.

Line 153. It is advisable to write down what types of RNAs lncRNAs can interact with.

Line 294. Please define the term “Tregs”.

Line 331: Check the grammar in the sentence.

Comments on the Quality of English Language

Moderate English editing required. The edit should address the use of singular and plural when describing lncRNAs. Often the authors write about a group of lncRNAs, but use the singular.

Author Response

Comments 1: The authors are encouraged to include more references to original research articles and reduce the number of references to reviews.

Response 1: Thank you for your comment. We have included more original researches and reduced the number of references to reviews (mainly in Introduction section).

Comments 2: Lines 162-173. To confirm the data presented in this paragraph, a reference to another review article was used. I would ask the authors to provide references to original research articles confirming the functions of lncRNAs for each designated function (i-viii).

Response 2: Thank you for your constructive comment. We have replaced the reference with some original research articles (in section “3. Biological characteristics and mechanisms of lncRNAs”).

Comments 3: Line 88. Probably a typo: BSCSs instead of BCSCs.

Response 3: Thank you for pointing this out. This sentence has been deleted due to the repetition throughout the manuscript.

Comments 4: Line 151. Better: Initially, lncRNA was considered as a byproduct of RNA polymerase II which did not have biological functions.

Response 4: Thank you for your suggestion. We have modified the sentence as you suggested.

Comments 5: Line 153. It is advisable to write down what types of RNAs lncRNAs can interact with.

Response 5: Thank you for your comment. We have added the type of RNA which can interact with lncRNA.

Comments 6: Line 294. Please define the term “Tregs”.

Response 6: Thank you for your comment. We have added the definition of the term “Trges”.

Comments 7: Line 331: Check the grammar in the sentence.

Response 7: Thank you for pointing this out. This sentence has been deleted due to the repetition throughout the manuscript.

Comments 8: Moderate English editing required. The edit should address the use of singular and plural when describing lncRNAs. Often the authors write about a group of lncRNAs, but use the singular.

Response 8: Thank you for pointing this out. We have corrected the singular and plural in describing lncRNAs (mainly in section “3. Biological characteristics and mechanisms of lncRNAs”).

Reviewer 3 Report

Comments and Suggestions for Authors

I have reviewed your manuscript entitled "Long non-coding RNAs as regulators for targeting breast cancer stem cells and tumor immune microenvironment: biological properties and therapeutic potential." While the overall content is promising, I would like to highlight some potential weaknesses and offer suggestions for improvement to enhance the clarity, relevance, and overall quality of your review article.

1. Introduction:

   The introduction provides a comprehensive overview of breast cancer subtypes, intratumor heterogeneity, and the cancer stem cell hypothesis. However, it lacks a sufficient introduction to long non-coding RNAs (lncRNAs), their fundamental mechanisms, and their established roles in cancer.

   Suggestion:

   Add more background information on lncRNAs, emphasizing their definition, major mechanisms, and emerging roles in cancer. This will better contextualize the importance of studying lncRNAs in relation to breast cancer heterogeneity and stemness.

2. Biological Properties Section:

   The section on lncRNA mechanisms lacks specific examples from the literature demonstrating these mechanisms in breast cancer stem cells and immune modulation.

  Suggestion:

   Enhance the descriptions of lncRNA mechanisms with 1-2 specific examples from relevant studies that showcase these mechanisms in the context of breast cancer stemness or immune regulation.

3. Organization and Flow:

   There is significant overlap between the Breast Cancer Stem Cells (BCSC) and Tumor Immune Microenvironment (TIME) sections, and the TIME section lacks a clear flow.

   Suggestion:

   Restructure the manuscript to integrate BCSC and TIME regulation seamlessly. Organize content logically under subheadings such as "Hypoxia" and "Stromal Cell Interactions" to improve clarity and flow.

4. Clinical Context:

   Some studies on lncRNAs appear preliminary or lack clinical validation, but this context is not sufficiently provided.

  Suggestion:

   Clearly indicate in relevant sections which lncRNA markers/mechanisms are in early preclinical stages and distinguish them from those with clinical validation. Provide context for prognostic studies, addressing limitations such as study size and reproducibility.

5. Repetition:

   There is noticeable repetition throughout the manuscript, contributing to excessive length.

  Suggestion:

   Streamline the manuscript by eliminating repetitive statements or redundant ideas that do not contribute new information. Focus on presenting only key essentials in the background and introduction.

I believe addressing these suggestions will significantly strengthen your manuscript. 

Comments on the Quality of English Language

manuscript demonstrates an acceptable level of writing quality. 

Author Response

Comments 1:

  1. Introduction:

   The introduction provides a comprehensive overview of breast cancer subtypes, intratumor heterogeneity, and the cancer stem cell hypothesis. However, it lacks a sufficient introduction to long non-coding RNAs (lncRNAs), their fundamental mechanisms, and their established roles in cancer.

   Suggestion:

   Add more background information on lncRNAs, emphasizing their definition, major mechanisms, and emerging roles in cancer. This will better contextualize the importance of studying lncRNAs in relation to breast cancer heterogeneity and stemness.

Response 1: Thank you for your comment. We have added more background information of lncRNAs in Introduction section.

Comments 2:

  1. Biological Properties Section:

   The section on lncRNA mechanisms lacks specific examples from the literature demonstrating these mechanisms in breast cancer stem cells and immune modulation.

  Suggestion:

   Enhance the descriptions of lncRNA mechanisms with 1-2 specific examples from relevant studies that showcase these mechanisms in the context of breast cancer stemness or immune regulation.

Response 2: Thank you for your comment. We have added some examples to show the mechanisms of lncRNAs (in section “3. Biological characteristics and mechanisms of lncRNAs”).

Comments 3:

  1. Organization and Flow:

   There is significant overlap between the Breast Cancer Stem Cells (BCSC) and Tumor Immune Microenvironment (TIME) sections, and the TIME section lacks a clear flow.

   Suggestion:

   Restructure the manuscript to integrate BCSC and TIME regulation seamlessly. Organize content logically under subheadings such as "Hypoxia" and "Stromal Cell Interactions" to improve clarity and flow.

Response 3: Thank you for your constructive comment. We have added the subheadings as you suggested to improve clarity and flow (in section “2.3. Role of TIME in CSCs”).

Comments 4:

  1. Clinical Context:

   Some studies on lncRNAs appear preliminary or lack clinical validation, but this context is not sufficiently provided.

  Suggestion:

   Clearly indicate in relevant sections which lncRNA markers/mechanisms are in early preclinical stages and distinguish them from those with clinical validation. Provide context for prognostic studies, addressing limitations such as study size and reproducibility.

Response 4: Thank you for your constructive comment. We have added some clinical context and limitations as you suggested (in section “6. Clinical implications of lncRNAs in BCSC and TIME”).

Comments 5:

  1. Repetition:

   There is noticeable repetition throughout the manuscript, contributing to excessive length.

  Suggestion:

   Streamline the manuscript by eliminating repetitive statements or redundant ideas that do not contribute new information. Focus on presenting only key essentials in the background and introduction.

Response 5: Thank you for pointing this out. We have properly deleted some content to reduce the repetition.

Comments 6:

manuscript demonstrates an acceptable level of writing quality.

Response 6: Thank you very much for the positive comments.

Round 2

Reviewer 2 Report

Comments and Suggestions for Authors

The authors responded to all comments and made the necessary changes.